# Research and Application Progress of Inverse Opal Photonic Crystals in Photocatalysis

**Hongming Xiang, Shu Yang \*, Emon Talukder, Chenyan Huang and Kaikai Chen**

School of Textiles and Fashion, Shanghai University of Engineering Science, Shanghai 201620, China; 19851543881@163.com (H.X.); emontalukder90@gmail.com (E.T.); 39200002@sues.edu.cn (K.C.)
* Correspondence: shuyang@sues.edu.cn

**Abstract:** In order to solve the problem of low photocatalytic efficiency in photocatalytic products, researchers proposed a method to use inverse opal photonic crystal structure in photocatalytic materials. This is due to a large specific surface area and a variety of optical properties of the inverse opal photonic crystal, which are great advantages in photocatalytic performance. In this paper, the photocatalytic principle and preparation methods of three-dimensional inverse opal photonic crystals are introduced, including the preparation of basic inverse opal photonic crystals and the photocatalytic modification of inverse opal photonic crystals, and then the application progresses of inverse opal photonic crystal photocatalyst in sewage purification, production of clean energy and waste gas treatment are introduced.

**Keywords:** inverse opal; photonic crystals; fabrication methods; photocatalytic application

## 1. Introduction

The world is seriously polluted and short of energy. Clean and renewable resources have attracted many researchers' attention and research. Solar energy is one of the most promising clean energy in the future, and photocatalyst is a material that can convert light energy into chemical energy, which plays an important role in the utilization of solar energy. Researchers have been committed to developing efficient and simple photocatalysts [1,2].

However, photocatalysts usually have a problem of low conversion. The introduction of inverse opal photonic crystal (IOPC) into the photocatalytic system has been proven to be a promising method to solve this problem [3–5]. Inverse opal photonic crystal is a certain kind of photonic crystal. Photonic crystal (PC) is a dielectric structure material with a photonic band gap, which is formed by materials with different dielectric constants and arranged periodically in space. It was independently proposed by John [6] and Yablonovitch [7] in 1987. Among the known photonic crystal structures, the three-dimensional photonic crystal formed by colloidal self-assembly has the same cubic close-packed structure as the natural opal structure, which is called opal photonic crystal; Another kind of three-dimensional ordered porous structure obtained by reverse replication of opal is called reverse opal photonic crystal. Inverse opal photonic crystals have many new physical properties and phenomena, such as slow photon effect [8], photonic band gap [9], photon localization [10–12], etc. Because of its periodic structure and excellent optical properties, inverse opal photonic crystals have been widely used in the field of photocatalysis [13–27].

In detail, the advantages of IOPC in photocatalytic systems are mainly based on the photonic band gap, slow light effect, and high specific surface area.

Photonic band gap: The inverse opal structure has a periodic refractive index and photonic band gap, which can significantly inhibit the propagation of light [28–31]. The periodic ordered porous structure of inverse opal structure makes it selective to incident light. A specific aperture can only allow the light of a specific wavelength to enter, and the incoming photons will be continuously reflected in the "restricted" structure, and the light

matching the electronic band gap of the photocatalyst will be absorbed by the photocatalyst material. Photons of other wavelengths cannot be absorbed by the catalyst because they do not match the electron band gap energy of the catalyst and will not affect the photon efficiency of the catalyst.

Slow light effect: The inverse opal structure can also produce structural dispersion and a slow light effect. At the wavelength corresponding to the stop band edge, photons propagate at a strongly reduced group velocity; Therefore, they are called "slow photons." If the energy of the slow photon overlaps with the absorptivity of the material, the photon absorptivity will increase with the increase of the effective optical path length. Curti et al. [32] studied the slow light effect of $TiO_2$ inverse opal photonic crystal photocatalysis. Their experimental results show that slow photons can enhance the absorption of materials; At the two edges of the stopband, the absorption rate of inverse opal increases sharply. At the red edge of the stopband, the photon absorption capacity of inverse opal is 2.7 times that of opal, while at the blue edge, the value is 1.6. Thomas et al. [33] also studied the slow light effect of inverse opal $TiO_2$ photonic crystal. They found that due to the slow light effect, the photocatalytic efficiency of the inverse opal photonic structure was seven times that of the opal photonic crystal structure.

High specific surface area: The ordered porous structure of inverse opal photonic crystal improves its specific surface area. The high specific surface area structure of inverse opal photonic crystal makes the active sites on the catalyst surface easier to be exposed, and photo-generated electrons can reach the active sites through the shortest migration path, so as to enhance the transfer of photo-generated charge and reduce the recombination of photo-generated electrons. Reducing the recombination efficiency of photo-generated charge holes is one of the methods to improve photocatalytic efficiency. Huang et al. [34] prepared phosphorus-doped inverse opal structure $C_3N_4$ and proved that the inverse opal structure could effectively separate the photo-generated charges and reduce the recombination rate of photo-generated charges.

This paper introduces the preparation methods of inverse opal photonic crystals, including the preparation of basic inverse opal photonic crystals and photocatalytic modification of inverse opal photonic crystals. And then, the application progresses of inverse opal photonic crystal photocatalyst in sewage purification, production of clean energy, and waste gas treatment are introduced.

## 2. Preparation of Inverse Opal Photonic Crystal

At present, a variety of methods have been developed for the preparation of inverse opal photonic crystals, including the template method, micromachining technology, multi-beam laser holographic pattern method, etc. [35,36]. This paper mainly introduces the most common preparation method of inverse opal photonic crystal, namely the template method. The template method can be divided into two-step and three-step methods [37].

The basic process of the two-step method is as follows: the first step is to disperse the colloidal microspheres in the precursor solution and self-assemble into a composite opal structure. The second step is to remove the microsphere template and obtain the inverse opal structure [38,39].

The process of the three-step method is shown in Figure 1. The specific steps are as follows: the first step is to build an opal photonic crystal template using colloid microsphere self-assembly; the second step is to fill the precursor in the resultant material and cure it; and the third step is to remove the template for the inverse opal structure.

Compared with these two methods, it can be seen that the two-step method has obvious limitations on the material, and only the material that can be directly obtained by heating the precursor can be used by the two-step method. So next, the three-step method will be introduced in detail.

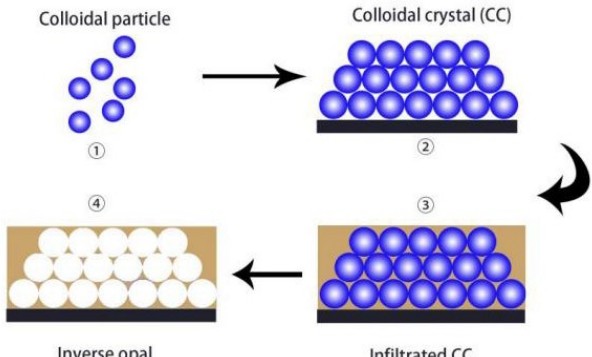

**Figure 1.** Preparation of inverse opal photonic crystals by three-step method.

### 2.1. Construction of the Opal Photonic Crystal Template

The construction of the opal template is the basis for the production of inverse opal photonic crystals, and the construction of the opal template includes nanolithography and the self-assembly of colloidal microspheres. Nanolithography, known as the "top-down" method, is expensive and slow. This paper mainly discusses the second method, which is the self-assembly of colloidal microspheres.

Self-assembly methods include the gravity sedimentation method [40], the vertical deposition method [41], etc. The gravity sedimentation method is to disperse the colloidal particle emulsion with uniform particle size and good monodispersity into the solvent according to a certain concentration. With the evaporation of the solvent, the colloidal particles are self-assembled on the material under the action of the gravity field to form a three-dimensional photonic crystal [42], as shown in Figure 2. The gravity sedimentation method has a simple preparation process and low requirements for equipment, but it has strict requirements for the size and density of colloidal particles, many sample defects, and a long preparation cycle. Gao [43] and others used silica nanoparticles (SNP) as materials to obtain silica photonic colloidal crystals via the self-assembly method through gravity sedimentation (Figure 3). In order to solve the problem of the long preparation period of the gravity sedimentation method, centrifugal force or thermal assistance can be introduced on the basis of the gravity sedimentation method to accelerate the deposition of colloidal particles [44]. The gravity sedimentation method also has many disadvantages, such as the long production time and low production efficiency. Because it only relies on the gravity of the ball itself, it is not suitable for large-scale industrial production.

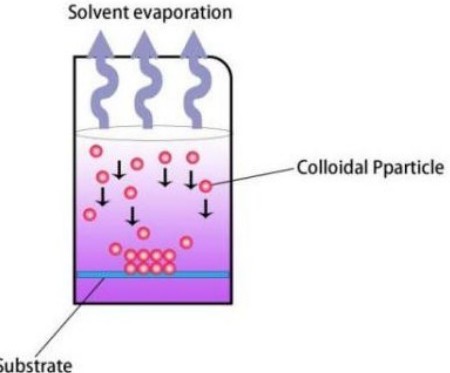

**Figure 2.** Preparation of inverse opal photonic crystals by gravity sedimentation method.

Vertical deposition is also a widely used self-assembly method. Figure 4 is a schematic diagram of the vertical deposition method; that is, the material is vertically placed in the assembly solution of monodisperse colloidal microspheres. With the evaporation of the solvent, colloidal microspheres gather on both sides of the material under the

combined action of capillary force and surface tension, forming a periodic photonic crystal structure [45]. Sinitskii A et al. [46] prepared a polystyrene colloidal opal template using the vertical deposition method, and prepared inverse opals based on different oxide materials ($TiO_2$, $SiO_2$, and $Fe_2O_3$). Compared with the gravity deposition method, the vertical deposition method has similar advantages and disadvantages, but it can make photonic crystals adhere to both sides of the material [47,48].

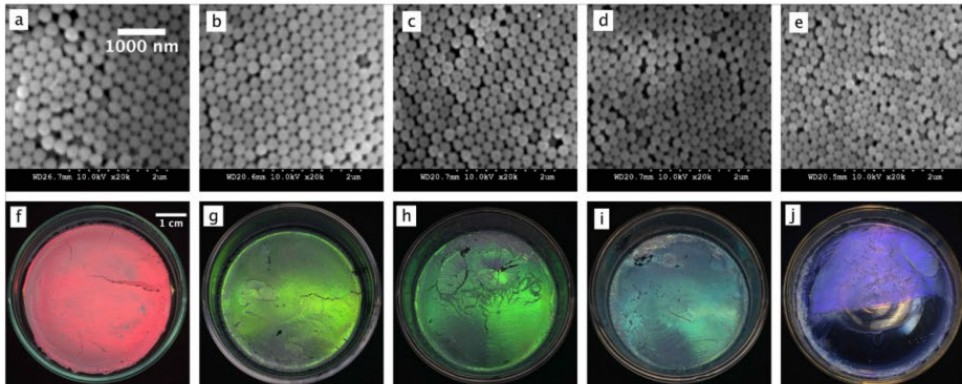

**Figure 3.** Top view SEM images (**a–e**) and images (**f–j**) of colored CC films with SNPs diameters of 350 nm, 282 nm, 270 nm, 249 nm, and 207 nm, respectively; scale bars are displayed in the first image of each set [43].

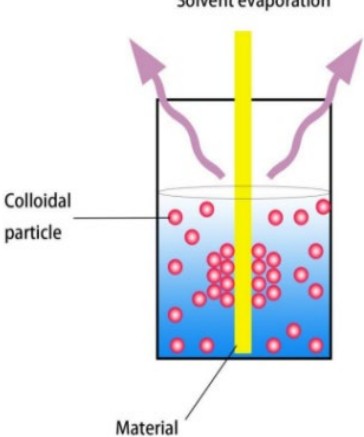

**Figure 4.** Schematic diagram of vertical deposition self-assembly method.

### 2.2. Filling of the Precursor

The additional material filling the opal template is called a precursor [49]; the precursor is crucial for the construction of inverse opal photonic crystals. To achieve inverse opal photonic crystals with an excellent morphological structure, the precursor must be uniformly filled in the opal template.

Precursor filling refers to the method of filling liquid precursor into the void of the opal template and curing via chemical reaction under specific conditions (Figure 5), including the sol-gel method [50,51], chemical vapor deposition method [52], atomic layer-by-layer deposition method [53], and electrochemical deposition method [54], etc.

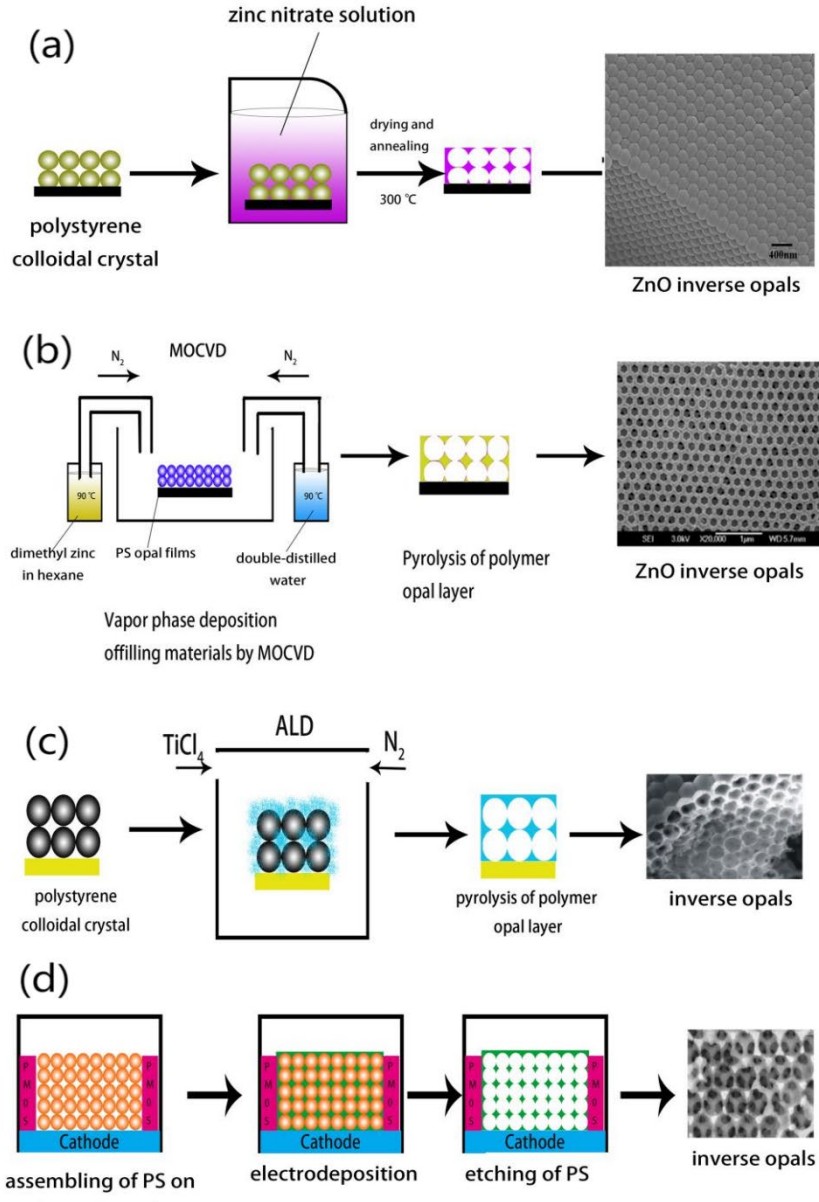

**Figure 5.** (**a**) Preparation of inverse opal photonic crystals by sol-gel procedure; (**b**) preparation of inverse opal photonic crystals by CVD method; (**c**) preparation of inverse opal photonic crystals by ALD method; and (**d**) preparation of inverse opal photonic crystals by electrodeposition method.

(1)  Sol-gel method: The sol-gel method uses hydrolyzing metal alkoxides and other precursors to fill the opal template under appropriate conditions, form a gel, and then calcine to obtain solid oxides. This method is suitable for the filling of most semiconductor oxide materials, but the filling rate is not high, and the volume shrinkage after drying is large. Some researchers have used the sol-gel method to prepare inverse opal photonic crystal photocatalysts [55–57]. Jie Yu [55] developed an inverse opal titanium dioxide photonic crystal photocatalyst to effectively degrade toluene. The catalyst was prepared via the sol-gel method using colloidal photonic crystal as a template. The catalyst was doped with carbon nitride quantum dots (CNQDs) in situ. The catalyst has good photocatalytic performance for toluene degradation. Under simulated sunlight irradiation, the samples were used to degrade the liquid pollutants represented by Rhodamine B (RhB) and phenol. As shown in Figure 6, the degradation rates of dyes and phenol reached more than 97% after 75 min and

100 min of illumination, respectively. Weijie Liu [57] improved the sol-gel permeation method. The prepared high-quality titanium dioxide inverse opal has a dense porous structure. In addition, they studied the optical properties and photocatalytic activity. The photocatalytic degradation of methyl orange was selected to evaluate the photocatalytic activity of the obtained titanium dioxide inverse opal. The conclusion is that compared with the samples prepared before improvement, the obtained titanium dioxide inverse opal has stronger photonic behavior and better photocatalytic activity.

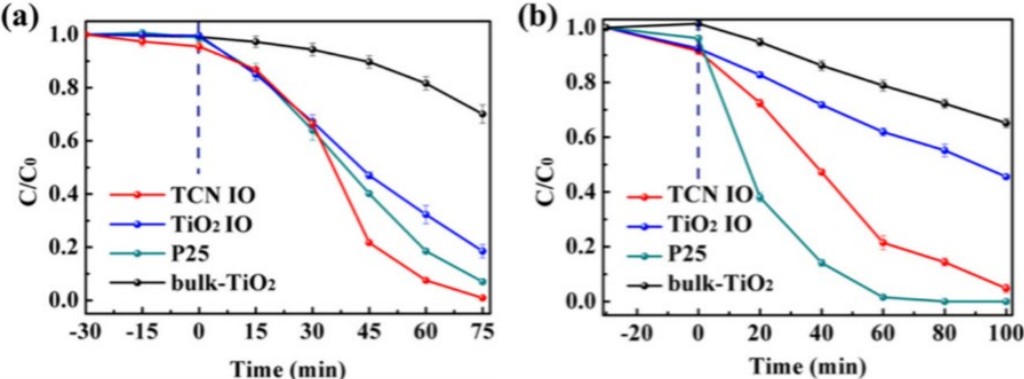

**Figure 6.** Photocatalytic degradation results in (**a**) 20 mg/L of RhB and (**b**) 10 mg/L phenol over the Nitrogen-doped titanium dioxide inverse opal photonic crystal (TCN IO) under simulated sunlight irradiation [55].

(2) The chemical vapor deposition method: can be used to diffuse the material to be filled in the form of gas precursor, adsorb it in the gap of the orderly microsphere, and then change the temperature or pressure to cause the precursor gas reaction, precipitate the solid material, and deposit it into the pores [58,59].

(3) The atomic layer-by-layer deposition method(ALD): this method is actually a form of chemical vapor deposition; it involves two or more kinds of vapor precursors on the solid surface and deposition to obtain the multilayer film method. The specific process is as follows. The surface or template is deposited in the gas phase of a certain amount of precursor, so that the surface reaches a single-layer saturated adsorption, and then the excess unabsorbed gas extraction is injected into another gas phase precursor. On the surface, two precursors are used to obtain a single-layer thickness film. Repeat the process to obtain a multilayer film with a specific thickness. Some researchers have used this method to prepare inverse opal photonic crystals with excellent photocatalytic performance. László Péter Bakos [60] used the atomic deposition method to fill the carbon nanosphere template to prepare the inverse opal photonic crystal. Their team previously showed that the carbon nanospheres (CS) were the appropriate template for the atomic layer deposition of $TiO_2$, because they were thermally stable at 300 °C in an inert atmosphere and had oxygen-containing functional groups on their surfaces. The hollow titanium dioxide shell can be prepared by subsequent annealing of the $CS-TiO_2$ composite. They used carbon nanospheres to prepare ordered face-centered colloidal crystals, and used ALD to deposit $TiO_2$ on them to produce inverse opal structure. They used scanning electron microscopy (SEM), Raman spectroscopy (Raman), X-ray diffraction (XRD), and UV-vis diffuse reflectance spectroscopy (UV-vis diffuse reflectance spectroscopy) to characterize the inverse opal samples, and investigated their photocatalytic degradation activity of methyl orange solution and methylene blue dye dried on the sample surface under UV and visible light irradiation.

Some researchers have also adopted an improved atomic deposition method to prepare inverse opal photonic crystals. Long [61] used $O_3$ as an oxidant to prepare inverse opal zinc oxide using the ALD method. Because $O_3$ has higher activity, this method has better

performance than the zinc oxide prepared via the $H_2O$ atomic deposition method and the electrodeposition method. This is because $O_3$ has higher activity and can produce oxide films with lower concentrations of impurities than films grown with $H_2O$. P. Birnal [62] synthesized $TiO_2$-Au composite inverse opal using the atomic deposition method. While depositing $TiO_2$, they also injected preformed Au nanoparticles. They compared the degradation of methylene blue (MB) by P-$TiO_2$-Au and IO-$TiO_2$-Au composite photocatalysts. The evolution of degradation percentage with time is shown in Figure 7. Two kinds of composite photocatalysts significantly improved the degradation rate of MB. The degradation rate of the P-$TiO_2$-Au flat film was 40% within 2 h of exposure and more than 90% within 14 h of exposure, while the degradation rate of the IO-$TiO_2$ photocatalyst was 95% only within 7 h of exposure to visible light. Their experiments show the potential of the atomic deposition method for preformed nanoparticles to produce complex composite structures.

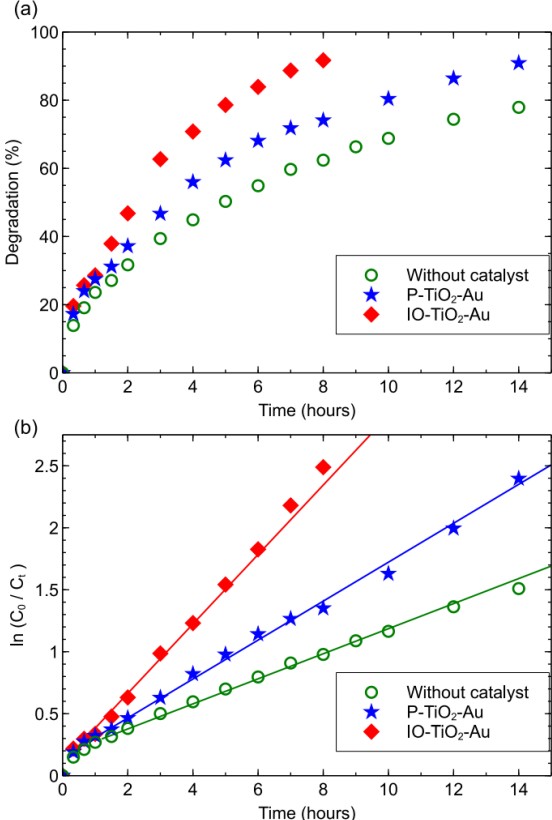

**Figure 7.** Degradation of methylene blue (1 μmol/L) over time under visible light. Illumination using P-$TiO_2$-Au and IO-$TiO_2$-Au films as photocatalysts, compared to the natural degradation of methylene blue. (**a**) Degradation percentage, (**b**) degradation kinetics in logarithmic scale [62].

(4) Electrochemical deposition method: the electrochemical reaction is used to fill the opal template directly by placing it directly in the cathode of the electrochemical battery. The electrochemical deposition method is characterized by the material filling continuously from the bottom to the top of the opal template, until the filling is relatively complete [63,64].

### 2.3. Removal of the Opal Template

The common template removal methods include dissolution and pyrolysis.

The dissolution method uses the chemical properties of the opal template material itself, and uses the chemical reagent dissolution to remove the opal template [47]. For example, the silica opal template can be removed using dilute hydrofluoric acid, and the polystyrene opal template can be removed with toluene.

The pyrolysis method is the high-temperature thermal decomposition using the physical properties of opal template material [57]. Min Wu [65] discussed the photocatalytic activity of photodegrading Rhodamine B (RhB) in calcined titania dioxide inverse opal films at different temperatures. The characterization of XRD, SEM, TEM, and HRTEM showed that the structures of inverse opal titanium dioxide photonic crystals were distorted with increasing temperatures. Excessive temperature will reduce the photocatalytic performance of inverse opal photonic crystals.

## 3. Modification of Photocatalysts for Inverse Opal Photonic Crystals

In order to further improve the photocatalytic efficiency of inverse opal photonic crystals, the researchers proposed the following modification methods, such as the metal modification method, the nonmetal modification method, the self-doping method, and others.

### 3.1. Metal Modification Method

The metal modification method involves loading the surface of nanomaterials via Au [66], Ag, Pt [67], Pd, and other precious metals, so as to improve the separation efficiency of electron-hole pairs. Some researchers also use metal compounds to improve the performance of inverse opal photonic crystal photocatalysts. Huang et al. [68] developed a new type of CuS-loaded inverse opal $g$-$C_3N_4$ photocatalyst (CN) to improve its photocatalytic activity for $CO_2$ reduction via the surface modification of CuS nanoparticles. In the experiment, inverse opal $g$-$C_3N_4$ with a good optical response and pore structure was prepared and characterized. Then, CuS nanoparticles were prepared by hydrothermal method and dispersed in toluene. Then, the CuS nanoparticles solution was added to the surface of inverse opal $g$-$C_3N_4$ and calcined at 110 °C for 1 h to obtain the CuS-modified inverse opal $g$-$C_3N_4$ photocatalyst. Finally, the photocatalytic reduction of $CO_2$ was carried out using the catalyst, and its photocatalytic performance and mechanism were studied. The CO generation rate is shown in Figure 8. The results showed that the optimal loading of CuS was 2 wt%, and the corresponding CO-evolution was 13.24 $\mu mol\cdot g^{-1}\cdot h^{-1}$, which is 3.2 times that of IO CN and five times that of bulk CN.

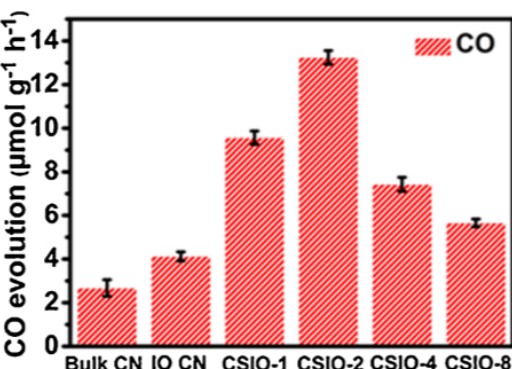

**Figure 8.** The CO production rate of the photocatalyst samples [68].

In conclusion, metal nanoparticles have a high light absorption rate and catalytic activity, which can effectively improve the photocatalytic efficiency of inverse opal photocrystals. In addition, by selecting different metal nanoparticles or changing the modification mode, the photocatalytic performance of inverse opal photocrystals can be optimized. Its modification method is simple, and its operation is relatively easy. However, it is worth noting that the stability of the metal nanoparticles is poor, and phenomena such as aggregation or dissolution may occur, which may affect the catalytic efficiency and lifetime. The matching between metal nanoparticles and inverse opal photocrystals should be considered in the preparation process to select the appropriate modification mode and conditions. Some metal nanoparticles may pose contamination and toxicity risks to the environment.

### 3.2. Nonmetal Modification Method

In order to develop an efficient inverse opal photonic crystal photocatalyst, the photocatalyst was modified by doping non-metallic elements such as C, P [69], N [70], and S on the structural basis of inverse opal photonic crystals. Some researchers have used a variety of non-metallic materials mixed with doping to improve the photocatalytic performance of inverse opal photonic crystals. Wenjun Zhang [71] prepared N-CD (nitrogen-doped carbon dot) via the one-step hydrothermal method and then used it to sensitize highly ordered porous $TiO_2$ IOS ($TiO_2$ inverse opals) films. Under simulated sunlight irradiation, the photocatalytic energy of N-CD/$TiO_2$ IOS and CD/$TiO_2$ IOS were compared. As shown in Figure 9, the photocatalytic degradation rate of MB (methyl blue) of the N-CD/$TiO_2$ IOS film also reached about 90% (curve 1), while the degradation rate of the CD/$TiO_2$ IOS film after five cycles was about 80% (curve 2). This fact illustrates that, in the photocatalytic degradation of MB molecules, the N-CD/$TiO_2$ IOS film has a higher catalytic performance and stability than the CD/$TiO_2$ IOS, because the n-doped atoms themselves have redox catalytic ability and the N-CD has a wider absorption range (UV and visible light regions).

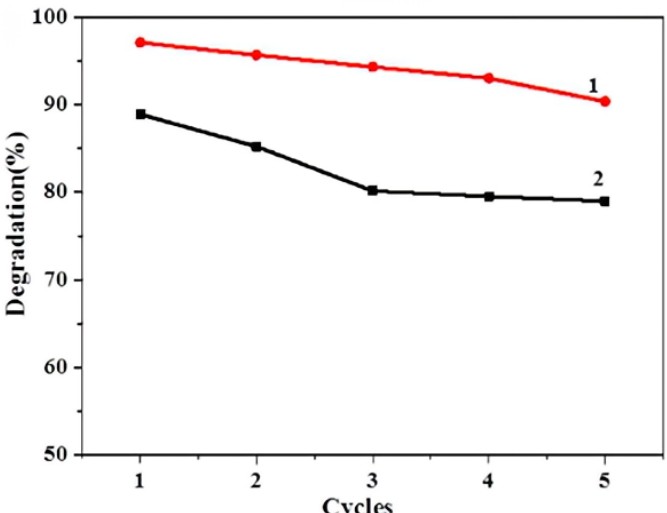

**Figure 9.** The degradation rate of MB by N-CD/$TiO_2$ IOS and CD/$TiO_2$ IOS.

### 3.3. Self-Doping Methods

Some researchers have utilized self-doping to improve the photocatalytic performance of inverse opal photonic crystals. Qi [72] studied a method to improve the photocatalytic performance of titanium dioxide ($TiO_2$). They self-doped $Ti^{3+}$ in a titanium dioxide inverse opal photonic crystal and used a slow-light effect to enhance the photocatalytic performance. Their research found that oxygen vacancies and $Ti^{3+}$ are able to narrow the band gap of $TiO_2$ and induce visible light capture. In order to study whether the synergistic effect of improving visible light absorption can be used to improve the photocatalytic activity driven by visible light, the photodegradation experiment of AO7 was carried out. The photocatalytic activity of pure $TiO_2$ was the lowest. The photocatalytic activity of doped $Ti^{3+}$ was improved after vacuum activation, which indicated that $Ti^{3+}$ and oxygen vacancy could indeed play a role in visible light photocatalytic activity. On the other hand, the T-170, T-265, and T-355 (170, 265, and 355, respectively, representing the concentrations of $Ti^{3+}$ in titanium dioxide inverse opal photonic crystals) samples showed similar photocatalytic activity. After vacuum activation, the band gap of titanium dioxide was narrowed due to the existence of $Ti^{3+}$ and oxygen vacancies. The inverse opal structure was able to improve the light trapping ability, generate photoelectrons and photoholes, and further improve the photocatalytic activity. V-T-355 (V represents vacuum activation) has the widest absorption and the highest photocatalytic activity in the visible light region. Therefore, the experiment

confirmed that the synergistic effect of inverse opals structure and vacuum activation on visible light capture could actually be used to improve photocatalytic activity.

### 3.4. Other Methods

In some studies, the inverse opal photocatalyst was combined with biomimetic chloroplast to improve the photocatalytic efficiency of inverse opal photonic crystal(Figure 10). Zhou et al. [73] took advantage of this characteristic and combined an inverse opal photonic crystal with bionic chloroplasts to improve its photocatalytic efficiency. Specifically, the researchers first prepared $TiO_2$ in an inverse opal template. The inverse opal structure, then the surface of the pore in the inverse opal structure, was loaded with chlorophyll (Chl) molecules and ionic liquid (IL) via the cation exchange method. Through the characterization and performance test of the prepared materials, the authors found that the material demonstrated good absorption properties in the UV-visible light region and displayed high photocatalytic activity. The $CO_2$ conversion rate reached 28.6%. In order to further improve the efficiency of photocatalysis, the author designed a "light-charge separation-transfer" mechanism based on the photosynthesis process in chloroplasts in nature. Under this mechanism, light energy was composed via $TiO_2$ in the inverse opal structure. The particles absorb and release the electrons, which were taken up by the chlorophyll loaded on the surface, where it was then efficiently excited into the biomass. The slow light effect in the inverse opal structure improved the light absorption and utilization efficiency of the material, enabling more photons to be absorbed and excite the charge, thus improving the photocatalytic efficiency. At the same time, the chlorophyll molecules in the inverse opal structure are able to form the biomimetic chloroplast structure, and the "light-charge separation-transfer" mechanism was adopted so that the light energy could be efficiently converted into biomass ($CH_4$), which further improved the photocatalytic efficiency. The slow light effect and biomimetic chloroplast structure in inverse opal structure are interrelated and act on the performance of the whole material. The slow light effect improves the light absorption and utilization efficiency, while the bionic chloroplast structure realizes the light-charge separation and efficient transformation.

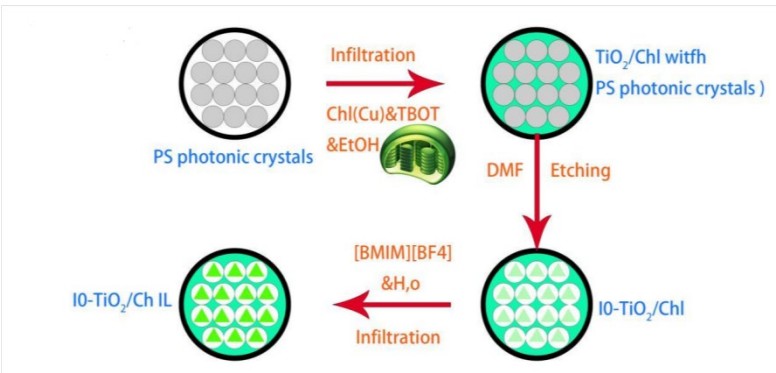

**Figure 10.** Preparation method of inverse opal photocatalyst combined with biomimetic chloroplast.

In addition to using biomimetic technology, some researchers utilized the heterojunction structure to improve the photocatalytic activity of inverse opal photonic crystals. The so-called heterojunction structure refers to the "S-type" barrier structure based on the built-in electric field. When the materials on both sides of the heterojunction have different conduction bands and valence band energy levels, the built-in electric field will be formed, so that the electrons and holes stimulated under light can move to both sides of the heterojunction, so as to realize effective charge separation. Moreover, the reverse photonic crystal structure can increase the light absorption efficiency and focus the photons near the interface through the photonic localized effect, further promoting the generation and transmission of photo-generated carriers. Therefore, the construction of heterojunctions enables efficient charge separation and rapid transport in photocatalytic reactions, thus

increasing the reaction efficiency. Liu et al. [74] studied a novel inverse opal photonic crystal $Bi_2WO_6$/$Bi_2O_3$ heterojunction photocatalyst with efficient charge separation and rapid migration to achieve highly active photocatalytic reactions. The results of this heterojunction via the sol-gel method showed that the photocatalysts demonstrated excellent photocatalytic performance under visible light and an obvious effect on grading the organic dye RhB.

## 4. The Application of Inverse Opal Photonic Crystals in the Field of Photocatalysis

As a photocatalyst, inverse opal photonic crystal is widely used in sewage treatment, clean energy production, and waste gas treatment, as shown in Figure 11.

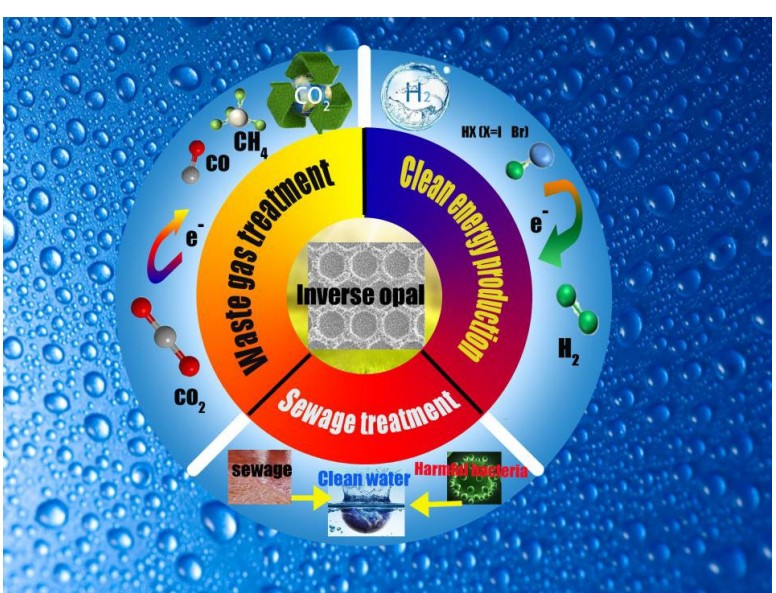

**Figure 11.** Application of inverse opal photocatalysis.

### 4.1. Sewage Treatment

In industrial production, industrial sewage treatment has always been a concern of researchers, who are making efforts to remove organic pollutants and antibiotic pollutants from the water. There are many existing treatment methods, such as filtration, adsorption, precipitation, photodegradation, and biodegradation [75]. Photodegradation, as a new clean and pollution-free organic pollutant treatment technology, has attracted the attention of researchers [76].

When studying water pollution, Rhodamine B (RhB) is generally used as a simulated pollutant. Because RhB is an artificially synthesized dye with a bright peach-red color that is easily soluble in water, it is widely used as a colorant in the textile and food industries and has a carcinogenic effect. Wastewater containing RhB would pollute the environment and harm human health and the growth of animals and plants.

The methods of the photocatalytic degradation of organic pollutants in water are as follows: (1) adsorption: pollutants are adsorbed on the surface of the photocatalyst; (2) electron excitation: due to the stimulation of external light source, when the photon energy obtained by the catalyst is greater than its own band gap, the electron will be excited from the valence band to the conduction band, so that a relatively stable hole will be left in the valence band, thus forming an electron-hole; and (3) redox reaction occurs. Photo-generated holes oxidize the $OH^-$ and $H_2O$ adsorbed on the surface of the catalyst into ·OH radicals with high activity and oxidation. Finally, ·OH oxidizes the organic pollutants adsorbed on the catalyst surface to $CO_2$ and $H_2O$ [77–79].

Inverse opal photonic crystals are widely used in the field of sewage treatment [80]. Wan et al. [81] successfully prepared an inverse opal $TiO_2$ photocatalyst which was able to react under visible light via the sol-gel method combined with an opal template. They used

the photocatalytic degradation of Rhodamine B to study the photocatalytic performance of the catalyst. They confirmed that the slow photon effect is able to enhance photocatalytic efficiency, and a large specific surface area can reduce the binding rate of photo-generated electron-hole pairs. Figure 12 shows the absorption spectra of RhB photocatalytic degradation and each self-photodegradation. In the process of photocatalytic degradation, the maximum absorption peak of RhB was weakened and blue-shifted. Because the photocatalyst almost completely degraded RhB in 1.5 h, the degradation rate was 92.5% after 1 h, indicating that the photocatalyst has a good photocatalytic performance.

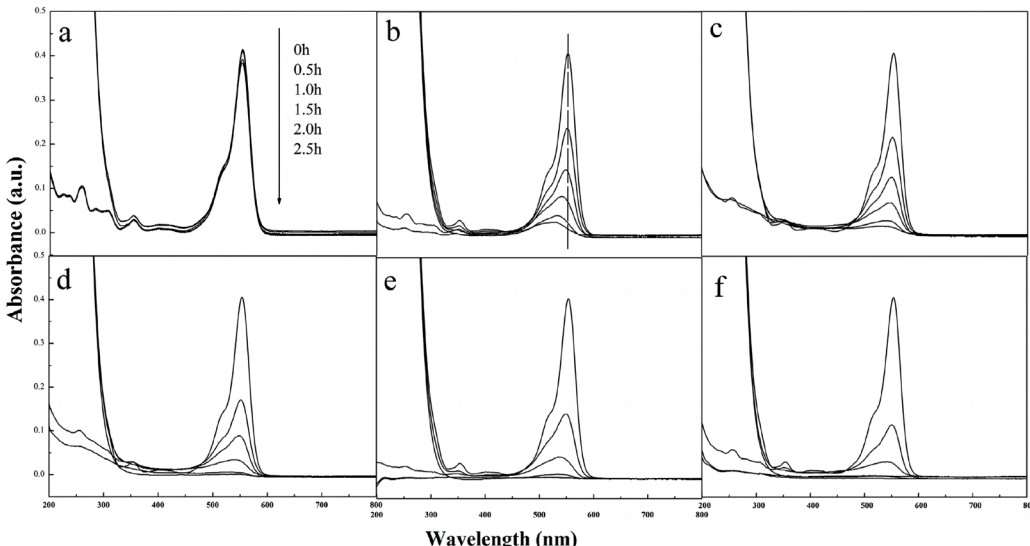

**Figure 12.** UV-vis absorption spectrum of RhB degraded by as-prepared samples under visible light irradiation; (**a**) self-degradation; (**b**) T-Sol; (**c**) IOT-230; (**d**) IOT-330; (**e**) IOT-440; (**f**) IOT-610 [81].

### 4.2. Clean Energy Production

As a kind of high-energy clean energy source, hydrogen is widely used in various industrial production activities. However, the synthesis process is complex and has high energy consumption, so how to use solar energy to produce hydrogen has become a concern of researchers. There are two main methods to produce hydrogen from solar energy.

Photoelectric water splitting method [82]: utilizing solar energy to generate electricity, which is used to electrolyze water molecules and decompose water into hydrogen and oxygen. Photocatalytic decomposition method [83]: using a photocatalyst to absorb the solar energy and generate a charge on its surface to start a reaction that decomposes the water into hydrogen and oxygen.

The photocatalytic decomposition method is relatively simple, and the use of photocatalytic decomposition to produce hydrogen does not require traditional energy consumption, and there is no emission of harmful gases such as carbon dioxide, making it an environmentally friendly method. Compared to the traditional electrolytic water method for hydrogen production, the use of photocatalysts can increase the reaction rate and have higher hydrogen production efficiency. And when using the photocatalytic decomposition method to produce hydrogen, the reaction conditions are mild, which can reduce the energy consumption and waste heat generated by the equipment, and improve the overall economy of the process. The most important thing is that the use of photocatalysts can achieve precise control of reaction rate and hydrogen production, which is of great significance for industrial production and scientific research. The principle of photocatalytic hydrogen production is that in the process of the photocatalytic decomposition of water, electrons in the conductive band reduce hydrogen ions to hydrogen, and holes in the valence band oxidize oxygen ions to oxygen.

Many researchers have conducted a lot of research on the preparation of hydrogen using inverse opal photonic crystal photocatalyst. Fiorenza et al. [84] prepared $TiO_2$-$BiVO_4$

and TiO$_2$-CuO samples with inverse opal structure, characterized them, and carried out photocatalytic water decomposition experiments under ultraviolet light and solar light irradiation. Figure 13 shows the H$_2$ production of the inverse opal titanium dioxide (I.O.TiO$_2$) system under UV and sunlight. Under UV irradiation, it can be seen that the hydrogen production rate of pure I.O.TiO$_2$ is higher than that of the TiO$_2$ photocatalyst purchased from the market (Figure 13A,B). In addition, it can be seen that the presence of BiVO$_4$ (Figure 13A) leads to a moderate increase in H$_2$ production. Under visible light irradiation, I.O.TiO$_2$ showed a five-time higher activity than the commercially available TiO$_2$ (Figure 13C,D). In this case, the I.O.TiO$_2$ 25% BiVO4 sample showed the best performance (purple line in Figure 13C). Lv et al. [85] prepared an efficient photocatalytic material called the inverse opal structure (IO)-TiO$_2$-MoO$_3$-x, which is able to catalyze H$_2$ generation and Rhodamine B dye degradation simultaneously. The material is made up of titanium dioxide (TiO$_2$), and the molybdate (MoO$_3$-x) composition has an inverse opal structure and plasma enhancement effect. The experimental results show that the photocatalytic activity of the material is not only seven times that of pure titanium dioxide photocatalyst but also has an effective catalytic effect in the visible light range. Liu et al. [86] identified an efficient photocatalyst, namely Ag (silver) modified onto g-C$_3$N$_4$ (melamine), an inverse opal photonic crystal structure, known as an Ag/g-C$_3$N$_4$ 3D inverse opal photonic crystal. The experimental results show that when using Ag-CN IO (Ag/g-C$_3$N$_4$ 3D inverse opal photonic crystal) as the photocatalyst, the H$_2$ release rate under ultraviolet light irradiation was 4.93 μmol·g$^{-1}$·h$^{-1}$, and because Ag nanoparticles enhance the optical absorption and the photonic crystal effect of the g-C$_3$N$_4$ inverse opal structure, the photocatalyst has excellent hydrogen production and stability.

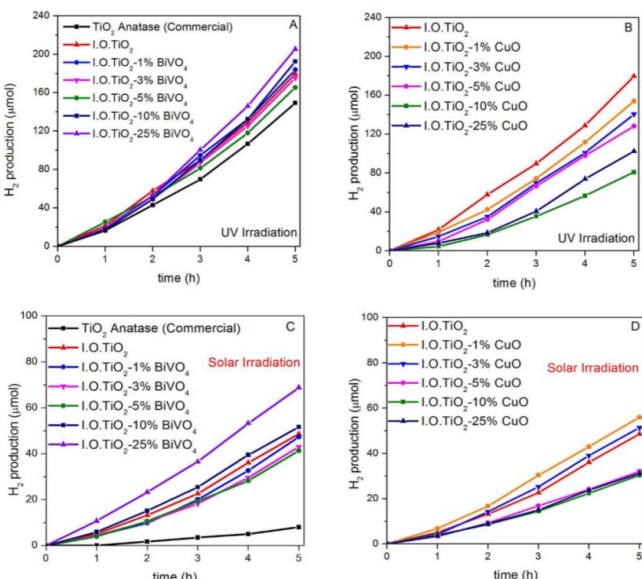

**Figure 13.** H$_2$ production of the inverse opal TiO$_2$ (I.O.TiO$_2$) system under UV and sunlight irradiation. (**A**) Inverse Opal TiO2-BiVO4 and (**B**)Inverse Opal TiO2-CuO composites under UV irradiation; (**C**) Inverse Opal TiO2-BiVO4 and (**D**) Inverse Opal TiO2-CuO under solar light irradiation [84].

### 4.3. Waste Gas Treatment

With the development of global industrialization, the emission of industrial waste gas has become one of the main factors endangering the earth's environment. As a waste gas that is widely emitted in industrial production, the decomposition of CO$_2$ has become a popular research direction among researchers. Converting carbon dioxide into organic compounds, such as methane or methanol, can both reduce the concentration of carbon dioxide in the atmosphere and also provide a new and convenient energy storage method [87]. In the conversion treatment of CO$_2$, due to its stable chemical properties, there are multiple problems, such as the low conversion efficiency of traditional catalysts and many reaction

by-products. Research has shown that photonic crystals exhibit high catalytic performance in the photo-reduction of $CO_2$.

Many researchers have conducted a great amount of research to degrade carbon dioxide by using inverse opal photonic crystal photocatalysts. Wherein Wei et al. [88] added Au nanoparticles to titanium dioxide inverse opal photonic crystal to prepare inverse opal photonic crystal with core-shell structure, and used it as a photocatalyst for $CO_2$ reduction reaction. The catalyst has the highest photocatalytic activity and $CO_2$ reduction selectivity, and the experimental results indicate that $CH_4$ generation rate of 41.6 $\mu$mol·$g^{-1}$·$h^{-1}$ and 98.6% selectivity for $CO_2$ reduction to generate $CH_4$. This composite has highly efficient photocatalytic $CO_2$ conversion properties under visible light. Xu et al. [89] prepared photocatalytic experiments to build rhenium doped into inverse opal $SnO_2$/$TiO_2$-x and reducing $CO_2$, and the $CO_2$ yield using the finally obtained catalyst was 16.59 $\mu$mol·$g^{-1}$·$h^{-1}$, which was about 1.21, 2.14 and 7.44 times obtained using inverse opal $SnO_2$/$TiO_2$-x, inverse opal $TiO_2$-x, and $SnO_2$, respectively.

Some examples of photocatalyst applications of the inverse opal structure are summarized in Table 1.

**Table 1.** Some examples of the photocatalytic applications of IOPCs.

| IOPC Materials | Photocatalysis Application | Result | Ref |
|---|---|---|---|
| Mg-$TiO_2$ | Sewage treatment | The Mg-$TiO_2$ system exhibits much higher activity than its counterpart due to the reduced band gap, which is due to the doping of $Mg^{2+}$ in the system. By adding $Mg^{2+}$, the sterilization rate under visible light can reach 100%. | [80] |
| g-$C_3N_4$-BiOBr | Sewage treatment | It provides a new idea for the preparation of a new visible light-driven Z-type photocatalyst and a new idea for the study of wastewater treatment methods. | [90] |
| $TiO_2$-$BiVO_4$ $TiO_2$-CuO | $H_2$ production | The synthesis of inverse opal material combined with $TiO_2$ structure modification and chemical modification through the addition of $BiVO_4$ or CuO can improve $H_2$ production via the photocatalytic decomposition of water. | [84] |
| $TiO_2$-$MoO_3$-x | $H_2$ production | The results show that compared with a single control factor, the composite of IO structure and plasma material ($MoO_3$) has higher light capture ability and carrier separation and transfer efficiency, which can significantly improve the photocatalytic activity of RhB degradation and hydrogen evolution. | [85] |
| Ag-$C_3N_4$ | $H_2$ production | The results showed that the hydrogen evolution performance of Ag-CN IO was the best among all the tested samples. | [86] |
| $TiO_2$-$ZrO_2$ | Carbon dioxide conversion | This study improved the oxidation-reduction ability of the material because the construction of inverse opal core-shell structure promoted the nano-crystallization of the material. | [55] |
| Au@CdS/IO-$TiO_2$ | Carbon dioxide conversion | Under simulated sunlight irradiation, the Au@CdS/IO-$TiO_2$ displayed excellent photocatalytic performance for the $CO_2$ reduction of $CH_4$. | [88] |
| $TiO_2$-x/$SnO_2$ | Carbon dioxide conversion | A strategy for constructing a new S-type heterojunction structure in visible light photocatalysts is proposed, which provides an ideal method for improving photocatalytic activity to treat organic pollutants and renewable energy production. | [89] |

## 5. Summary

The inverse opal photonic crystal, because of its own excellent characteristics—large surface area, interrelated channel structure, more active catalytic sites, quicker electron transmission speeds, a slow photonic effect able to enhance the absorption of light, good mass transfer properties, and a three-dimensional order within a multiple scattering structure, has great research value for improving photocatalytic efficiency.

However, there are still many challenges in the application of inverse opal photonic crystals in photocatalysis. For example, inverse opal photonic crystals cannot be produced on a large scale, their service life is not long enough, and the photocatalytic efficiency of inverse opal photonic crystals still needs to be improved. In the near future, researchers may find solutions to these problems and make significant progress in the application of inverse opal photonic crystals in the field of photocatalysis.

**Author Contributions:** Conceptualization, S.Y.; writing—original draft preparation, H.X.; writing—review and editing, E.T.; visualization, C.H.; funding acquisition, K.C. All authors have read and agreed to the published version of the manuscript.

**Funding:** This research was funded by the Natural Science Foundation of Zhejiang Province [LY20E030009], the National Natural Science Foundation of China [51403078 and 52103035], China Scholarship Council [201508330017].

**Institutional Review Board Statement:** Not applicable.

**Informed Consent Statement:** Not applicable.

**Data Availability Statement:** Not applicable.

**Conflicts of Interest:** The authors declare no conflict of interest.

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
