# Peer review of "Research and Application Progress of Inverse Opal Photonic Crystals in Photocatalysis"

_inorganics, doi:10.3390/inorganics11080337_

Round 1
Reviewer 1 Report (New Reviewer)
In the manuscript "Research and Application Progress of inverse opal Photonic Crystals in Photocatalysis" the authors review the preparation, properties and use of inverse opal structures in the field of photocatalysis. The review is hard to read and needs major revision before it can be published.
In the following i give single examples of problems that need to be addressed. It does not provide an exhaustive list of things that need to be changed as there are similar problems throughout the review which need to be addressed. The authors should carefully revise their manuscript and have it perhaps be checked by an independend reader to check if the readability has been sufficiently improved.
In general: You provide a good survey on all the research that has been done in the field, but you do not clearly point out what has been great advancements and what appear to be the most promising approaches.
The english needs to be improved (see below).
You need to be more careful with your statements without providing evidence for them: "Clean and renewable resources have received the attention and research of the majority of researchers. Solar energy is the most promising clean energy in the future.". How do you know that majority of resaerchers are active in research on clean and renewable resourses.? - if you write of "many reasearchers" it would be o.k. - How do you know that solar energy is the most promising? - if you write is a "highly" promising... it would be o.k.
All abbreviations should be introduced at their first use and it would be helpful not to use abbreviations in the figure captions.
All texts and all stuctures should be readable or clearly visible in the figures. Text in Fig.5 you can not read without zooming in as well as the structure in the first SEM.
TZS powder: What is TZS, sample 156 IOP-S@TZS: What is the meaning of this nomenclature.
Make the figure captions better understandable without reading the text. E.g. Fig. 6 Provide some information on the samples that were tested.
"Under the same experimental conditions, two kinds of titanium dioxide inverse 179 opals were prepared before and after improvement". You do not specify what kind of improvement was used.
Even with looking at Fig. 5b it is not clear what kind of structrue is produced according to your description: "Juarez B H et al.[49] used a dilution solution of dimethyl zinc in hexane as a zinc precursor to grow ZnO in PS opal films on silicon support. They used double steam water (DSW) as the oxygen precursor and put the silicon hydride into the void of the opal template through low pressure chemical vapor deposition. In this method, the two reaction sources were stored separately in a glass bubble apparatus at 90℃. With nitrogen as the carrier gas, the precursor is continuously fed into the reactor through a steady flow rate. This method utilizes the ability of the pregas to physically penetrate the pores of photonic crystals, resulting in a dense inverse opal structure." Is this a mixed oxide, or multilayer or what?
"Pyrolysis is to use the physical properties of opal template material by high temperature melting." Confusing: Pyrolysis is typically not just melting but completely decomposing a material.
Typo: 10,00 should be 1000
"photocatalytic properties at the incidence angles of 20°, 45°and 0°, respectively". For what are thes incidence angles? Why is there a dependence?
"the light wave will be modulated to produce a band gap, and defects will be generated in the inverse opal structure" I do not understand how light modulation generates defects. Please explain.
Once again: Please write the review in a way that a person not knowing the original papers is able to follow what you write. Check it by an independend reader.
The english needs to be improved. Please have it be checked by a native speaker.
Example: the article "the" is used if you refer to a "known" subject otherwise you need to use "a". Already in the abstract:
researchers have proposed a scheme (because you have not mentioned a specific scheme you are now refering to).
various optical properties of a photonic crystal (you are not to a specific photonic crystals but to photonic crystals in general).
"the surface or template to be deposited in the gas phase of a certain amount of precursor, 226 so that the surface reaches a single layer saturated adsorption; then the excess unab-227 sorbed gas extraction, another gas phase precursor injected into the surface, on the sur-228 face two precursors to get a single layer thickness film; repeat the process can get a mul-229 tilayer film with a certain thickness."
I do not get the meaning of this text
"In this work, we... " It does not seem you refer to your own work.
what is "atomic deposition method" do you mean "atomic layer deposition method"
"Moreover, the long--range ordered TiO2 was obtained by modifying the long--range samples porous membrane. " I do not understand what you mean.
Author Response
Please see the attachment.

Reviewer 2 Report (New Reviewer)
Review Inorganics-2512200 for the Authors: In this work, the Authors browse through inverse opal photonic crystals for catalytic application. Straight from the Abstract, the text seems to be introduction style and not abstract style. Something has been said about inverse opal photonic crystals, something about preparation and something about application but all quite diffuse and unclear. Structure and microstructure have been vaguely correlated, preparation methods are not even specified and application will be introduced? The abstract has to be deleted and written from the start. In introduction, the authors could have done better when explaining inverse opal photonic crystals. When addressing preparation, much has been said about sort of parent methods, and that is not necessary. I guess that it is clear that plenty of methods are suitable for some sort of templating, do we have to name that all. I would rather hear more about specific inverse opal crystal related preparing details. There is no shift between preparing and photocatalytic performance reports. Why? Then back to another group of preparing and then back to photocatalytic properties. Then modifying, then doping, then other. The structuring of the paper is obviously really bad. In addition, I think that segmentation into 2step, 3-step and other preparing routes is not really relevant. Application section is actually not bad, but there are issues such as: Is really RhB relevant for sewage degradation study? In addition, text on CO2 and H2 reports should have been broader. Conclusion are just generic. The language is ok, the graphics are ok. Overall, the paper deals with interesting materials but fails to put the matter into a neatly structured, interesting paper worth publishing. Unfortunately I must suggest reject.
Fine.
Author Response
Dear Reviewer,
Thank you for your kind suggestions, we have revised our manuscript according to your advice. We also provide a point-by-point response, please see the attachment.
Thanks again for your valuable time, and looking forward to receiving your further advice.
Yours,
Shu Yang

Reviewer 3 Report (New Reviewer)
The manuscript is a review of the literature devoted to the three-dimensional inverse opal photonic crystals suitable for use in photocatalysis. It introduces the preparation methods of material of such structure, the principle of photocatalysis, and the advantages of inverse opal photonic crystals in the field of photocatalysis, as well as the modification methods to further improve the efficiency of photocatalysis.
The subject is interesting, particularly from an applied point of view. Although some discussions of the influence of structure on photocatalytic efficiency and photocatalysis itself are vague, the manuscript gives valuable information that could be useful for readers.
However, the manuscript is carelessly written and before my recommendation for publication, many technical weaknesses should be removed. Some of them are listed below.
Fig 2 All letters should be equal size; Pparticle
Fig 5 All letters should be equal size (may be as in 5b)
148 What are RhB, TZS, IOP-S@TZS ?
237 “In this work, we prepared ordered face centered colloidal crystals”
We?
248 “ H2O.As shown in Figure 9,”
H2O, as shown in Figure 9.
I do not understand what proves the Fig 9.
299 “adopted calcination temperatures of 550,700,800,900 and 10,00℃ respectively”
10,00 or 1000oC?
338 “photonic band gap [59-60], That is, within a certain”
, that is, … ?
347 “material; For”
; for ?
343 “A specific aperture can only reflect light of a specific wavelength, The light of other wavelengths will enter the inverse opal structure, and the photons entering it will be constantly reflected in the inverse opal structure as if they were "confined", and the light matching with the electronic band gap of the photocatalyst will be absorbed by the 346 photocatalyst material; For photons of other wavelengths, because they do not match the 347 electronic band gap energy of the catalyst, they cannot be absorbed by the catalyst and 348 will not affect the photon efficiency of the catalys”
Should be reformulated.
367 “nanoparticles to transfer slow light to BiVO4”
What is “slow light”?
368 “by controlling transmission speed of photons”
What is “transmission speed of photons”?
571 “when the photocatalyst photonic energy is greater than its gap width, the valence band in the electron will be excited to the belt, leaving relatively stable hole on the valence band, thus forming electron-hole.”
Should be reformulated.
578 “Finally, · Oh oxidizes the organic pollutants”
Something is missing.
Author Response
Please see the attachment.

Reviewer 4 Report (New Reviewer)
The Authors submitted a review on the inverse opal photonic crystals and their application for photocatalysis. The review describes various approaches to template deposition (sol-gel, CVD, ALD, electrochemical deposition) and template removal. I strongly appreciate the mutual comparison of various results; hence, the submitted review is not just a summary but offers added value. As a result, I support the manuscript in its present form.
No comments on English level.
Author Response
Dear Reviewer,
Thank you for your time and kind suggestions.
Round 2
Reviewer 1 Report (New Reviewer)
The manuscript has been much improved and is now much better understandable and valuable for a broader readership.
Author Response
Dear Reviewer,
Thank you for your time and kind suggestions.
Reviewer 2 Report (New Reviewer)
Review Inorganics-2512200-r3 for the Authors: In the revised work dealing with inverse opal photonic crystals for catalytic application, the Authors introduced few more references and commented these results, but that hardly help the case of main problem, and that is relevant structure of the paper. Namely, the quite diffuse and unclear frame of the paper has not been affected. Nothing has been done to correlate microstructure to preparing routines; actually, the paper just browses through the preparing routines without any apparent guideline. Furthermore, I really miss a link between very crude simple dyes and sewage style application. While some efforts are there, I still feel the paper is not at the level suitable for publishing. Unfortunately, I must suggest reject.
Minor corrections necessary.
This manuscript is a resubmission of an earlier submission. The following is a list of the peer review reports and author responses from that submission.
Round 1
Reviewer 1 Report
Review article of H. Xiang with co-authors is devoted to synthesis and photocatalytic applications of inverse opal photonic crystals. Review’s merit is a large number of publications over the past 5 years: 43 of 77 publications in the list of references. The selected review topic is relevant, but not new (for example, see publications [Inverse Opal Photonic Crystals as a Strategy to Improve Photocatalysis: Underexplored Questions. M. Curti et al. J. Phys. Chem. Lett. 2015, 6, 19, 3903–3910; Engineered inverse opal structured semiconductors for solar light-driven environmental catalysis. J. Gao et al. Nanoscale 2022, 39, 14, 14341–14367]). This review lists the results of a number of articles without their systematization; there are no general tables and conclusions. In the Introduction section, there is no logic in presenting the material (see, for example, paragraph 2, lines 29–52). In the Introduction, there is information only about titanium oxide, other semiconductors are not mentioned. However, the advantages and disadvantages of titanium oxide authors are extended to all semiconductors. It is not clear what the authors mean by "traditional TiO2 material products"? It is not clear how the synthesis of complex structures from spherical colloids (see lines 142–154) relates to the section “1.1.1. Construction of the opal photonic crystal template”? The review contains too much disputed and incorrect statements without any references (for example, see lines 354–374). I do not recommend this review for publication in Inorganics.
Reviewer 2 Report
The review entitled „Research and Application Progress of inverse opal Photonic”
Crystals in Photocatalysis“ by Xiang et al. aims to give an overview of the synthesis of inverse opaline photonic crystals with a special application focus on photocatalysis.
Overall, I, unfortunately, must admit that the review manuscript is of poor quality, unstructured, incomplete, and cannot be accepted at all in the current version.
I believe the topic is of general interest, but a clear structure and a much deeper overview have to be given. The number of references with 77 is only a portion of the field; hence, I suggest much broader and deeper literature research about a topic that includes preparation and application.
Comments
The introduction starts by putting the context in an overall context and then depicting a specific material, i.e., TiO2, which is known to be photocatalytic active, and explaining the disadvantages, specifically the position of the semiconductor’s bandgap in the UV. However, the authors list other disadvantages, which all need to be justified by references (and maybe even shortly explained – p1 l39-41:
· Low photocatalytic efficiency: On the one hand, the claim TiO2 is good, on the other hand, the state low photocatalytic efficiency (so compared to what ?), ref missing
· Small specific surface area: this is not the disadvantage of the material but rather of the structure of the material; ref missing.
· Small adsorption capacity: again, compared to what? goes along with the surface area
· Easy occurrence of electron-hole recombination: Ref missing
· Low Raman enhancement factor: why is this important for photocatalysis, Ref missing.
Afterward, other more general drawbacks for semiconductors are listed, but without any reference, such as photo corrosion, water and hydrogen production, bandgap determining the absorption, low photocatalytic efficiency, and fast recombination rate, to name a few. BUT all without any reference.
Then they state that one of the problems, namely the photocatalytic efficiency, can be solved, by using photonic crystals. They directly jump the slow light/photon effect without explaining it much and state that this helps to improve the photon absorption efficiency. In the current form of the introduction, the authors wildly jump between the different topics and mix them up. Here, a much better ordering is needed to take the reader at hand all of course backed up with references:
· Statement: photonic crystals can improve the phototcatalytic efficiency
· What is a photonic crystal, which examples can be given nanoporous gold, modulated anodic alumina or titania, direct and inverse opals, etc. – here only structure plays a role (key word: periodic arrangement of dielectric constant/refractive index)
· What is the consequence out of the refractive index modulation? Answer a photonic stopband/bandgap.
· What is the consequence of the photonic stopband? Answer: slow photon/light effect.
· What is the slow photon effect?
· How does the slow photon effect contribute to photocatalytic efficiency? Answer: interaction probability with medium/semiconductor increases in photonic crystals.
· Other advantages: if porous structures are used, surface area is higher.
1. Preparation
· l108 ref missing for vertical convective self-assembly method.
· Why is gravity sedimentation method the “most concise” one? Personal opinion or ref?
· l117: “Structural color” is given by the sizes of the different spheres used. But more interestingly, and this is the origin, the photonic stopband can be controlled.
· L125 several refs for vertical deposition method needed, if this is the “most widely used” technique.
· Image missing for “novel photonic crystals with complex structure” -l153
· Fig5: SEM images not clearly visible, the colloidal crystal in sketch is bcc and not fcc. Is this really a bcc crystal?
· For (1 – sol gel): Several materials examples are given. Why and how do they contribute to photocatalysis? The focus should be given to the preparation of photocatalytic active photonic crystals.
· 2 – chemical precipitation: Why no materials examples?
· Fig 9 again bcc crystal in sketch. Fig 9 also seems to be stretched.
· L228 bold line in manuscript
· Statement l245-l247 about temperature tolerance and surface groups for ALD is not true at all. There are tons of publications out where ALD has been used for coating polymeric particles and especially polymeric opals.
· “and the preparation effect is not ideal.” For DEZ plus water. What does that mean? DEZ plus water is one of the gold processes in ALD. So what do you mean by performance?
· Ref missing for “light stability”
· L296-308 phases can be also stabilized by doping into titania. (l298 1000 °C instead of 10,000 °C)
I was rather unclear about which material examples have been chosen. I agree that one has to decide which to highlight, but similar to the Introduction, it seemed to me rather arbitrary and unstructured.
From now on, I just highlight the main issues of the review, and not all the small mistakes and incorrectness presented, but basically, every subchapter has to be proven and checked for logic order.
2.1. should go into the introduction
2.2. is a mix-up of everything, and also it seems that there was a conceptual misunderstanding what the physical differences between opals and inverse opals are.
· Also direct opals show an ordered porous structure: statement l355/356 is wrong.
· Light is not absorbed at all in a photonic crystal (besides the materials absorption): light is reflected in a certain wavelength region because of the photonic stopband. The rest of the light is transmitted through the opal and can then just be absorbed by the material the opal is made from.
· Statement: “inverse opal can produce structural dispersion” this is also true for direct opals
· Fig 12: absorption spectra of opal and reference plate shown.
· Fig14: again bcc instead of fcc
· Several effects play a role by introducing metallic nanoparticles into an inverse opal which have to be separately explained, e.g., plasmon resonances, charge carrier separation, etc.
· k Fig 15 has to be specified when used the first time (maybe in the introduction). To make clear what is the physical quantity which can be measured and compared.
· Fig 16 What is the information given? The focus of the review should be photocatalytic activity, not color.
· Overall I am missing information, graphs, and tables on how the structure and the materials are improving the photocatalytic activity.
· Discussion about the alignment of the photonic stopband/bandgap in relation to the semiconductors bandgap or dye absorption band is totally missing or at least hidden. Red edge/blue edge how do they have to be aligned so that an increase can be expected by making use of a photonic crystal?
· Ref missing for statements in l569-574
· “Novel honeycomb…” In what sense are they novel? Is it just structure, then it should go in 2. If it is an application then this should be highlighted and should make clear why (and in what quantity) honeycomb outperforms fcc inverse opals.
· Fig 19: This is a rather complex structure to start within that field. Before it should be answered, how does a TiO2 inverse opal compare to a direct TiO2 opal? Then the next step is following
· What is LVX?
· Avoid using words, such as good, high, and large, to name a few, without giving a reference value. For example l619 “showed good photocatalytic hydrogen production performance”. What mean good? What is the baseline or the target value? Good compared to what? Another example l669 “This composite has highly efficient photocatalytic….” compared to non-functionalized inverse opal!
· …..
I highly encourage the authors to rewrite and overthink the complete manuscript: give it an order, define criteria for which references should be highlighted and which should just be cited, e.g., novel approach, novel materials, significantly improved photocatalytic activity compared to reference samples,… Only 3 (maybe 4) out of 21 figures show really photocatalytic data, the rest are schemes and mainly SEM images which are somehow eye-catching, but two or three are sufficient, and the rest should deal with the influence of material and structure on the photocatalytic activity.
Tables for different materials, structures, etc, should be created since they are comprehensive summary of the different topics.
The English should be carefully revised. Since I reject the manuscript, I will not list all single mistakes and errors.
But just check for:
- sentences without verbs,
- wording (material, structure, products, and devices all have different meanings and cannot be randomly used,
-consistencies: e.g., band gap vs bandgap,
-photon vs photonic
-colloid vs colloidal